

# Evaluating the utility of camera traps in field studies of predation

Christopher K. Akcali[1,2], Hibraim Adán Pérez-Mendoza[3],
David Salazar-Valenzuela[4], David W. Kikuchi[5], Juan M. Guayasamin[6]
and David W. Pfennig[1]

[1] Department of Biology, University of North Carolina at Chapel Hill, Chapel Hill, NC, USA
[2] North Carolina Museum of Natural Sciences, Raleigh, NC, USA
[3] Facultad de Estudios Superiores Iztacala, Universidad Nacional Autónoma de México, Tlalneplanta, Mexico
[4] Facultad de Ciencias de Medio Ambiente, Universidad Tecnológica Indoamérica, Quito, Ecuador
[5] Department of Ecology and Evolutionary Biology, University of Arizona, Tucson, AZ, USA
[6] Colegio de Ciencias Biológicas y Ambientales, Universidad San Francisco de Quito, Quito, Ecuador

Corresponding author
Christopher K. Akcali,
akcali@live.unc.edu

## ABSTRACT

Artificial prey techniques—wherein synthetic replicas of real organisms are placed in natural habitats—are widely used to study predation in the field. We investigated the extent to which videography could provide additional information to such studies. As a part of studies on aposematism and mimicry of coral snakes (*Micrurus*) and their mimics, observational data from 109 artificial snake prey were collected from video-recording camera traps in three locations in the Americas (*terra firme* forest, Tiputini Biodiversity Station, Ecuador; premontane wet forest, Nahá Reserve, Mexico; longleaf pine forest, Southeastern Coastal Plain, North Carolina, USA). During 1,536 camera days, a total of 268 observations of 20 putative snake predator species were recorded in the vicinity of artificial prey. Predators were observed to detect artificial prey 52 times, but only 21 attacks were recorded. Mammals were the most commonly recorded group of predators near replicas (243) and were responsible for most detections (48) and attacks (20). There was no difference between avian or mammalian predators in their probability of detecting replicas nor in their probability of attacking replicas after detecting them. Bite and beak marks left on clay replicas registered a higher ratio of avian:mammalian attacks than videos registered. Approximately 61.5% of artificial prey monitored with cameras remained undetected by predators throughout the duration of the experiments. Observational data collected from videos could provide more robust inferences on the relative fitness of different prey phenotypes, predator behavior, and the relative contribution of different predator species to selection on prey. However, we estimate that the level of predator activity necessary for the benefit of additional information that videos provide to be worth their financial costs is achieved in fewer than 20% of published artificial prey studies. Although we suggest future predation studies employing artificial prey to consider using videography as a tool to inspire new, more focused inquiry, the investment in camera traps is unlikely to be worth the expense for most artificial prey studies until the cost:benefit ratio decreases.

## INTRODUCTION

Studies of predator-prey interactions are often difficult since natural predation events are challenging to observe (*Irschick & Reznick, 2009*). Moreover, the ability of the rare observation of single predation events to provide general insights into predator-prey interactions is inherently limited. To overcome both difficulties, artificial replicas of prey species are commonly used to study predation in the wild. Such facsimiles allow key features of prey phenotypes (e.g., color, pattern, shape, or size) to be easily manipulated and produced in large numbers, thereby allowing predation to be studied in diverse natural populations (*Irschick & Reznick, 2009*). Generally, these studies involve constructing replicas (e.g., of pre-colored, nontoxic clay) bearing different colors, patterns, and shapes and placing several hundred of these replicas in natural habitats, where they are exposed to predation by naturally occurring, free-ranging predators. After a pre-determined period of time, each replica is scored as attacked or not based on the number and type of marks left on it. Conclusions are then drawn based on the patterns of attacks across phenotypes and/or habitats. Such artificial prey techniques have been used to address a wide variety of evolutionary and ecological questions, ranging from predator psychology to aposematism and mimicry (reviewed in *Bateman, Fleming & Wolfe (2017)*). These studies have been used to measure predator-mediated natural selection on diverse taxa, including insects (*Lövei & Ferrante, 2017*), fish (*Caley & Schluter, 2003*), frogs (*Saporito et al., 2007*), salamanders (*Kuchta, 2005*), turtles (*Marchand et al., 2002*), lizards (*Stuart-Fox et al., 2003*), snakes (*Pfennig, Harcombe & Pfennig, 2001*), birds (*Ibáñez-Álamo et al., 2015*), and mice (*Vignieri, Larson & Hoekstra, 2010*).

This traditional approach of using replicas to study predation in the field has three major shortcomings. First, predation attempts—and the identity of the predators—are inferred (*Irschick & Reznick, 2009*). Although most marks left by predators permit broad classification of predator type (e.g., beak imprints indicate avian predation), they rarely permit predators to be identified to species (*Irschick & Reznick, 2009*). Furthermore, replicas can be easily removed by predators, making it impossible to determine if predation even occurred. Second, only a subset of interactions between replicas and predators can be assessed from marks left on replicas (*Irschick & Reznick, 2009*). For example, predators might detect the replicas and decide not to attack them (*Willink et al., 2014*). Most studies consider all replicas not bearing attack marks as equivalent in statistical analyses, but a variety of factors can affect the probabilities that predators detect a replica as well as not attack a replica after detecting it. Third, replicas are unlikely to sample all potential predators (*Irschick & Reznick, 2009*). Predators that rely heavily on movement (e.g., felids) or smell (e.g., canids) to detect prey might ignore motionless or odorless replicas (*Irschick & Reznick, 2009*). In sum, new and improved insight into predation could be gained from artificial prey studies if additional information on the identity and behavior of predator species could be collected.

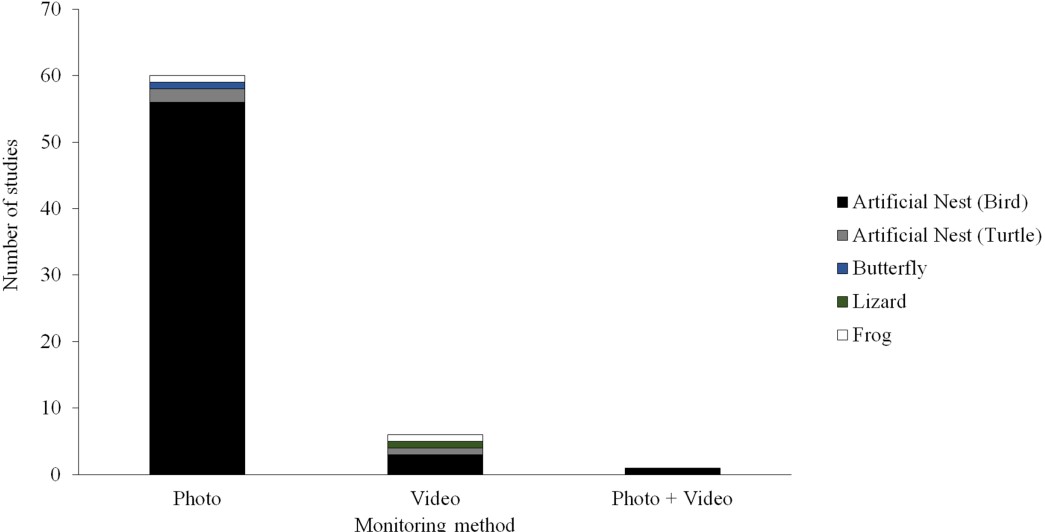

**Figure 1 Field studies of predation.** Number of field studies of predation employing camera traps using different types of monitoring methods and different types of artificial prey. Manuscripts were informally searched in Google Scholar (http://scholar.google.com) using a variety of search terms (e.g., artificial prey, artificial nest, clay model, and predation) and taxon terms (e.g., amphibian, bird, butterfly, frog, lizard, salamander, and snake). The search was conducted December 23, 2017.

Camera trapping technology could provide a potentially useful tool for field studies of predation. A camera trap consists of a remotely activated camera that is equipped with a motion or an infrared sensor (some also use a light beam as a trigger). This technology has been used in ecological research for decades (*Savidge & Seibert, 1988*; *Griffiths & Van Schalk, 1993*; *O'Connell, Nichols & Karanth, 2011*; *Burton et al., 2015*), typically to detect or survey the abundance of naturally occurring animals. Although several field studies of predation have experimented with camera trapping techniques, most of these studies have used still images to monitor predator activity (*Picman, 1987*; *Paluh, Kenison & Saporito, 2015*; *Ho et al., 2016*; *Hanmer, Thomas & Fellowes, 2017*) and only a few have used video (*Thompson & Burhans, 2004*; *Latif, Heath & Rotenberry, 2012*; *Sato et al., 2014*; *Willink et al., 2014*; *Jedlikowski, Brzezinski & Chibowski, 2015*; *Dziadzio et al., 2016*; Fig. 1). Most these studies using video to monitor predator activity near artificial prey have been conducted on small spatial scales (e.g., at one or a few sites with similar habitat) and have only used videos to aid the identification of predators attacking prey.

Here, we studied the ability of camera trap videos to provide additional information to field studies of predation employing artificial prey. The "prey" in our studies are highly venomous New World coral snakes and various similarly patterned harmless species, which are aposematic and mimetic prey, respectively, bearing conspicuous phenotypes that have long been thought to facilitate the evolution of avoidance behaviors in predators (*Bates, 1862*; *Smith, 1975*, *1977*; Fig. 2). We used camera traps to extract observational data from three independent artificial prey field experiments (*Akcali, Kikuchi & Pfennig, 2018*; Supplementary Data). We did so to quantify the frequency at

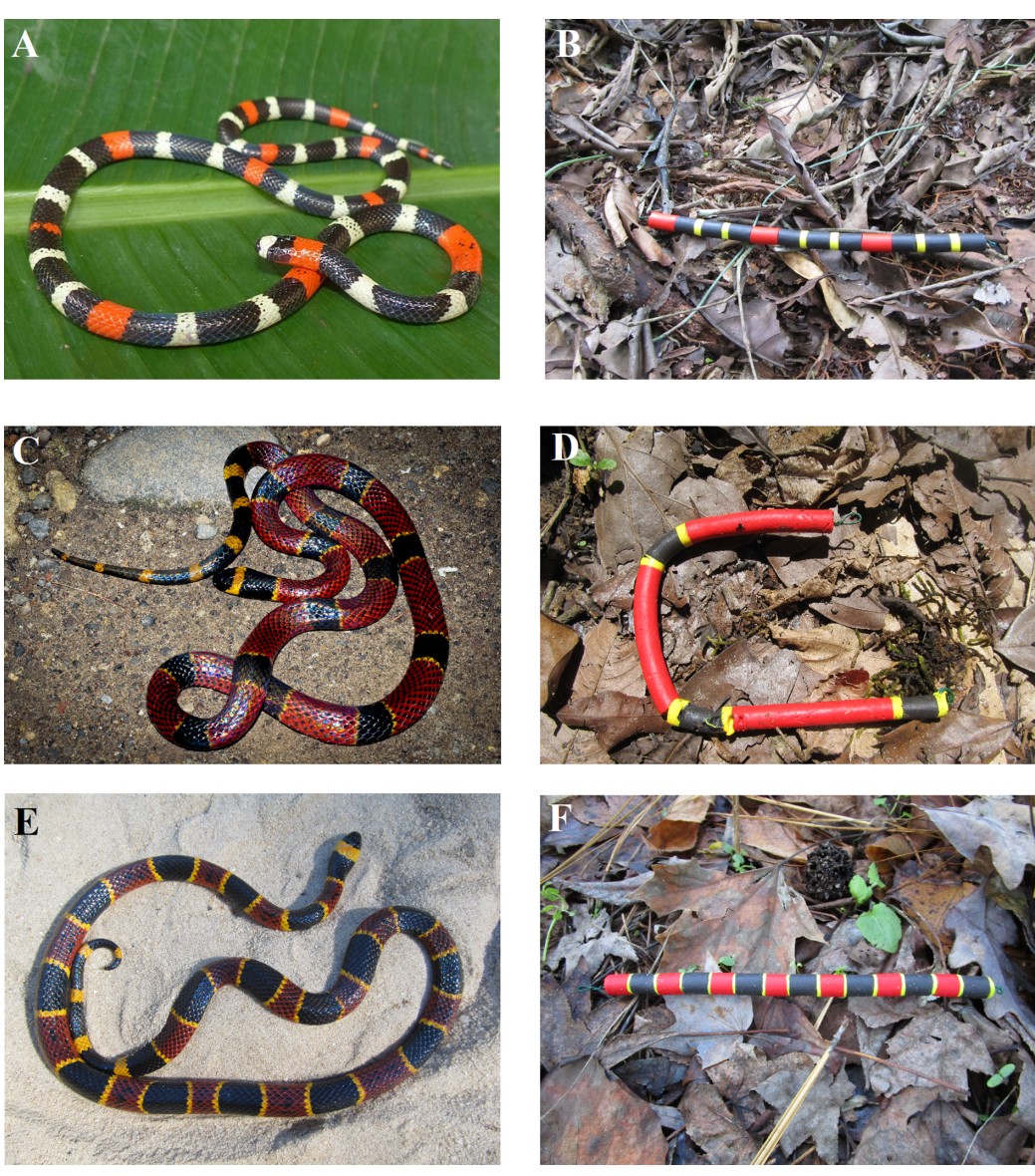

**Figure 2 Study snake species.** A sampling of images of live snakes (A) (C) and (E) and artificial snake replicas (B) (D) (F) from each experimental location. (A, B) The South American coral snake (*Micrurus lemniscatus*) (photo credit: Mike Pingleton), (C, D) the variable coral snake (*Micrurus diastema*) (photo credit: Eric Centenero Alcalá), and (E, F) the eastern coral snake (*Micrurus fulvius*) (photo credit: Christopher K. Akcali). Note the bite marks and change in shape caused by a mammalian predation attempt in D.

which predators encounter, detect, and attack artificial prey. Using these data, we asked the following questions. First, what are the relative frequencies at which predators encounter, detect, and attack replicas? Second, how do the frequency of encounters, detections, and attacks by predators vary temporally? Third, how does predator type, avian vs. mammal, affect the probability that predators detect and attack artificial prey? Fourth, how does the frequency at which predators encounter, detect, and attack prey vary between predator species? Fifth, how do clay marks and videos differ in their ability to register avian vs. mammalian predation attempts? After answering these questions, we conclude

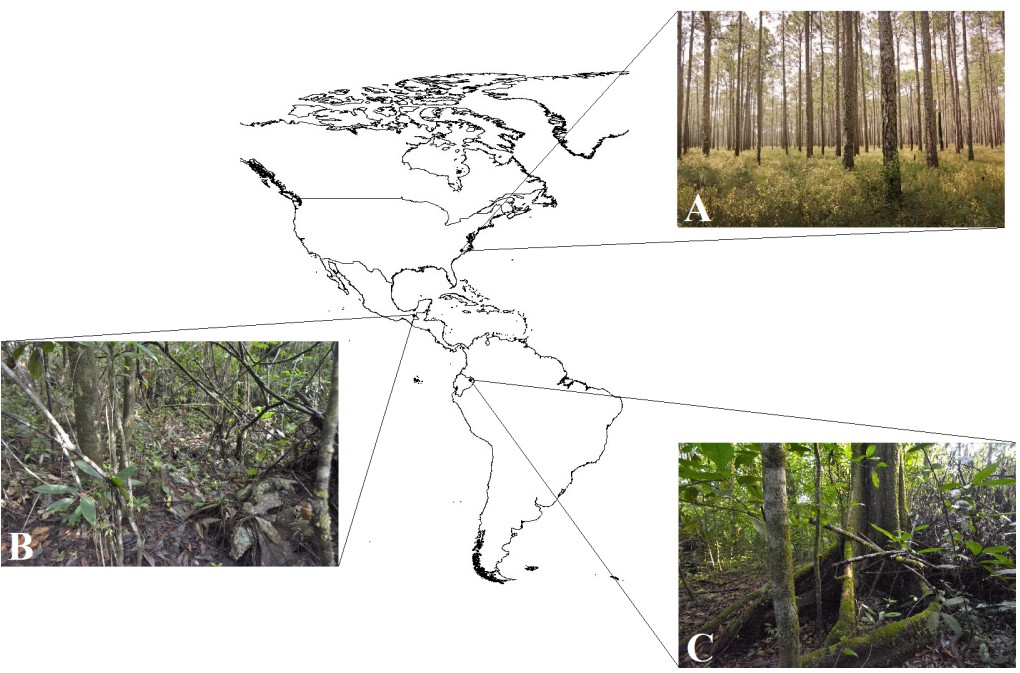

**Figure 3 Study areas.** Camera traps were used to collect observational data on predator behavior in three field experiments, conducted in North Carolina, USA, Mexico, and Ecuador, that were aimed to test hypotheses of aposematism and mimicry. Insets show habitat typical of the study areas: (A) longleaf pine forest, North Carolina, USA (Photo Credit: Christopher K. Akcali); (B) premontane wet forest, Chiapas, Mexico (Photo Credit: Christopher K. Akcali); and (C) *terra firme* rainforest, Orellana, Ecuador (Photo Credit: Christopher K. Akcali).

by discussing some of the costs and benefits of incorporating videography into field studies of predation.

## MATERIALS AND METHODS

### Ethics statement

Data collection used non-invasive, remotely-triggered camera traps and hence did not involve direct contact or interaction with animals. The clay used in all experiments is nontoxic. Fieldwork was done under the following permits: Ecuador–N° 002-017 IC-FAU-DNB/MA; Mexico–SGPAJDGVS/09347/16. No permits were required in North Carolina, USA.

### Field experiments

Three field experiments using clay replicas of various species of coral snakes and their presumed mimics (Fig. 2; Table S1) were conducted at three separate locations in the Americas (Fig. 3). The first experiment was conducted in February 2017 in Amazonian lowland rainforest at Tiputini Biodiversity Station, Orellana, Ecuador (~0°37′S, 76°10′W, 190–270 m asl; Table 1). This experiment is a part of a larger study that seeks to understand the causes of diversity in aposematism. In this experiment specifically, the aim was to characterize the pattern of selection on a set of aposematic phenotypes in a region where coral snake diversity is high. The second experiment was conducted from

**Table 1 Field experiments.**

| | Ecuador | Mexico | North Carolina, USA |
|---|---|---|---|
| Number of phenotypes | 5 (four *Micrurus* variants + brown control) | 4 (three *P. elapoides* variants + brown control) | 3 (three *M. fulvius* variants) |
| Length of replicas | 165 mm | 250 mm | 180 mm |
| Number of transects | 27 | 35 | 20 |
| Minimum distance between transects | 200 m | 200 m | Three km |
| Placement of replicas in transects | Singly, along forest trails, and one to four m off trails on alternating sides | Singly, along forest trails, and one to four m off trails on alternating sides | Each variant in groups of three off trails; all replicas attached to nails |
| Distance between replicas or sets of replicas | 5–10 m | 5–10 m | 50–75 m |
| Replicas with cameras | 37 | 22 | 69 |
| Replicas without cameras | 1,313 | 1,378 | 531 |
| Days replicas without cameras left in field | 6 | 12 | 28 |
| Days replicas with cameras left in field | 6, 8, or 14 | 30 | 28 |
| Replica days | 8,356 | 17,196 | 16,800 |
| Interval replicas were checked | 2 days | 6 days | Replicas not checked during experiment |

Note:
List of characteristics of field experiments that aimed to test hypotheses of aposematism and mimicry in Ecuador, Mexico, and North Carolina, USA. Camera traps were employed at a subset of replicas to collect observational data on predator activity near artificial prey replicas.

June to July 2017 in premontane wet rainforest at Nahá Reserve, Municipality of Ocosingo, Chiapas, México (~16°58′N, 91°35′W, 800–1,200 m asl; Table 1). The goal of this experiment was to test the "multiple models hypothesis" of imprecise mimicry, which proposes that mimics might evolve imprecise mimicry as a consequence of a selective trade-off to resemble multiple model species (*Edmunds, 2000*). The third experiment was conducted from October to November 2017 in longleaf pine forests of the Sandhills and Coastal Plain of North Carolina, USA (~34°45′N, 78°32′W, 0–150 m asl; Table 1). This experiment was a part of a larger study that tested whether a coral snake species and its mimics were engaged in a coevolutionary arms race (*Akcali, Kikuchi & Pfennig, 2018*).

Clay replicas in all experiments were constructed using pre-colored, odorless, nontoxic Sculpey III modeling clay. Measurements of preserved snake specimens from several museums (see the specific museum collections listed in Appendix S1) and photographs of live specimens were used to design prey phenotypes in each experiment. Replicas in all experiments were one cm in diameter, but varied in length (Table 1). Because each field experiment was a part of its own independent study, the experiments varied in several ways (Table 1). All damaged replicas were replaced with new replicas during each experiment if transects where checked before their designated date of retrieval (Table 1). Sampling effort for each field experiment in terms of replica days was calculated by multiplying the number of days that replicas were left in the field by the total number of replicas that were placed in the field. The latter includes the number of replicas in front of
cameras (regardless as to whether the camera was functional or not) as well as the number of replicas without cameras.

## Camera trapping

We used several relatively inexpensive (<USD $100) digital camera traps (Spypoint Force 10, Scout Guard SG560V-31B, ANNKE C303, Bestguarder DTC-880V) triggered by an infrared motion-and-heat detector to obtain observational data on predator activity near replicas during each field experiment. Cameras used a variable number of AA batteries and were equipped with 32-gigabyte SD cards. In each experiment, we attached cameras to the trunks of nearby trees and positioned them ~0.75–1 m above the surface of the ground at an approximately 45° downward angle. In Ecuador and Mexico, cameras were placed randomly among transects, approximately one meter away from single replicas and were set to have a high sensitivity (if sensitivity could be altered). In North Carolina, cameras were placed approximately two to three m in front of sets of three replicas in a clustered fashion (i.e., cameras were placed at every set of replicas in two transects and part of a third transect) and were set to have a medium sensitivity. Average distances between cameras were 1.25 ± 0.817 km, 1.37 ± 0.829 km, and 4.60 ± 4.11 km in Ecuador, Mexico, and North Carolina, respectively. Although vegetation that, when blown by wind, might falsely trigger the cameras was cleared prior to arming the cameras, we tended to place cameras at sites that were devoid of such vegetation to minimize disturbance to the habitat. Cameras were programmed to take 60 s videos when triggered. Videos were associated with data on the location (from GPS), identity of the camera, date, and time. All data collected from camera traps were recorded using data standards developed for the use of camera traps in biodiversity research (*Forrester et al., 2016*).

Sampling effort for each field experiment in terms of camera days was calculated by taking the sum of the total number of days that each camera was functional in the field. In Ecuador, we placed 27 camera traps (13 Spypoint; 10 Scout Guard; one ANNKE) in front of replicas for 14 days. Five camera traps (five Spypoint) were placed in front of replicas for 8 days and then moved in front of replicas in other transects for the final 6 days. Three cameras (three Scout Guard) failed to take video throughout the duration of the field experiment and one camera (one Spypoint) took video for 10 days until a spider built a dense web in front of the lens, making it impossible to make out any animal activity on video thereafter. Thus, cameras in Ecuador were armed for a total of 402 camera days ((23 cameras × 14 days) + (1 camera × 10 days) + (5 cameras × 8 days) + (5 cameras × 6 days)). In Mexico, we placed 22 camera traps (21 Spypoint; one ANNKE) in front of replicas for 30 days. One camera (one ANNKE) failed to take video throughout the duration of the field experiment. Thus, 21 cameras in Mexico were armed for a total of 630 camera days (21 cameras × 30 days). In North Carolina, we placed 23 cameras (21 Spypoint; one ANNKE; one Bestguarder) in front of replicas for 28 days. Five cameras (four Spypoint and one ANNKE) failed to take video throughout the duration of the field experiment. Thus, 18 cameras in North Carolina were armed for a total of 504 camera days (18 cameras × 28 days). In Ecuador and Mexico, replicas in front

of cameras were often exposed to predation longer than replicas that were not monitored by cameras (Table 1).

## Analyses

All vertebrate species that triggered the cameras were recorded. Although a variety of vertebrate species have been documented to prey on coral snakes and their mimics, including frogs, toads, snakes, caimans, hawks, falcons, kestrels, shrikes, anis, puffbirds, skunks, and mustelids (*Roze, 1996*; *Campbell & Lamar, 2004*), we focus on potential avian and mammalian predators in this study as reptiles and amphibians were rarely detected on cameras and would likely not be selective agents for aposematic coloration. Furthermore, we excluded potential rodents and lagomorph predators from analyses, as has often been done in previous studies (*Brodie, 1993*; *Kikuchi & Pfennig, 2010*), as well as non-predatory passerines, doves, and tinamou species, as these species would likely not represent significant threats to real snakes (see list of vertebrate species considered as predators in analyses in Table S1). Although our choice of which species to consider as predators might be inaccurate, our focus in this study is on the ability of camera traps to provide additional information. So although we refer to all species captured on videos that might be snake predators as "predators" throughout the manuscript out of convenience, we recognize that it would be more appropriate to refer to many of these predator species as "potential predators."

We noted whether each video demonstrated an encounter, detection, attack, and avoidance by a predator. Encounters were simply defined as videos that contained a predator. However, we classified videos of predators as belonging to independent encounters if more than 30 min had elapsed between consecutive videos of the same species at the same location. We used 30 min as a cut-off because visits by herds of peccaries (*Tayassu pecari* and *Peccari tajacu*) were typically the longest of any species at any given site among the three experimental locations, but most visits were less than 30 min. Thus, when we use the term "videos," we are referring to the unit (i.e., the actual number of videos) that cameras have taken. In contrast, when we use the term "encounter," we are referring to independent records of predator presence that might include several videos. Detections were defined as encounters where a predator clearly detected a replica (i.e., the predator decreased the rapidity of its movement near the replica and directed attention toward the replica either with its eyes or nose). Attacks were defined as detections where a predator bit a replica (Video S1–S7). Avoidances were defined as detections that did not result in an attack (Video S8–S10). Obviously, cases of avoidance may have arisen because a predator failed to recognize a detected replica as a snake but made a decision not to attack. Thus, when we use avoid, we do not make the implicit assumption that predators recognize replicas as snakes.

Prior to reviewing camera records, all replicas with and without associated camera traps were scored in the field as attacked or not attacked, based on the presence or absence of tooth and beak marks, or missing (i.e., no trace of the replica was present). At each replica or sets of replicas with cameras, we then tallied the number of encounters, detections, and attacks by predator species using camera trap videos. We classified

predator activity and behavior by hour, starting at midnight, to examine diurnal patterns. Diurnal activity and behavioral patterns were sufficiently well marked that statistical tests were not needed. We also asked how likely predators were to detect a replica they had encountered, and to attack a replica they had detected. We modeled the probability that a predator would detect a replica given that it had encountered it—that is, P(Detection|Encounter) and the probability that a detection would result in an attack—that is, P(Attack|Detection). To obtain a sample size sufficient for analysis, we pooled data across Ecuador and Mexico to analyze P(Detection|Encounter), and across Ecuador, Mexico, and North Carolina to analyze P(Attack|Detection). We used different datasets for these two analyses because in North Carolina, cameras were directed at triads of replicas rather than individual replicas, making the calculation of P(Detection|Encounter) different from that in Ecuador and Mexico. We used the glmer function in the lme4 package to fit logistic regressions of whether or not each encountered replica was detected (or attacked, in the second model) as a function of whether the predator was a bird or a mammal, with transect and replica identity included as random effects. Analyses at the species level were not possible due to the low sample sizes of individual species.

We also asked whether there was a difference in detecting attacks by birds vs. mammals using marks left in clay or videos. We tested whether the proportion of attacks by birds vs. mammals differed between clay marks and videos using Fisher's exact test.

## RESULTS

### Predator activity patterns

After eliminating videos with no identifiable animal or only with people, we had 1,071 videos. After classifying videos not separated by at least 30 min per species at a given site as representing single records, we had 906 encounters. After eliminating encounters by species that were not classified as snake predators, we were left with 268 encounters of 20 predator species (Table 2), which included 25 encounters of six avian predator species (six families; Table 2) and 243 encounters of 14 mammalian predator species (eight families; Table 2).

Across all experimental locations, we found no difference between avian and mammalian predators in their probability of detecting replicas after encounter in Ecuador and Mexico (Fig. 4; Likelihood ratio test; $\chi_1^2 = 0.2$; $p = 0.79$). We found no difference between avian and mammalian predators in their probability of attacking replicas after detection in Ecuador, Mexico, and North Carolina (Fig. 4; Likelihood ratio test; $\chi_1^2 = 0.01$; $p = 0.92$). In total, videos captured 21 attacks and 31 avoidances (Table 3).

The frequency of encounters increased approximately five and 12 times more rapidly than the frequency of detections and attacks, respectively, as a function of camera trapping effort (Fig. S1). The frequency of detections increased approximately 2.4 times more rapidly than the frequency of attacks (Fig. S1).

The timing of encounters, detections, and attacks varied among experimental locations (Fig. 5). In Ecuador, activity peaked during daylight hours (Fig. 5). In contrast, in

**Table 2 Predator species.**

**Ecuador**

| Family | Common Name (Scientific Name) | Encounters | Detections | Attacks |
|---|---|---|---|---|
| Bucconidae | Brown nunlet (*Nonnula brunnea*) | 2 | | |
| Tayassuidae | Collared peccary (*Peccari tajacu*) | 26 | 11 | |
| Dasypodidae | Giant armadillo (*Priodontes maximus*) | 1 | | |
| Psophiidae | Gray-winged trumpeter (*Psophia crepitans*) | 16 | 4 | 1 |
| Dasypodidae | Nine-banded armadillo (*Dasypus novemcinctus*) | 6 | 2 | |
| Felidae | Ocelot (*Leopardus pardalis*) | 2 | | |
| Tayassuidae | Peccary sp. | 8 | | |
| Accipitridae | Slate-colored hawk (*Buteogallus schistaceus*) | 2 | | |
| Tayassuidae | White-lipped peccary (*Tayassu pacari*) | 4 | 1 | 1 |
| | Total | 67 | 18 | 2 |

**Mexico**

| Family | Common Name (Scientific Name) | Encounters | Detections | Attacks |
|---|---|---|---|---|
| Didelphidae | Common opossum (*Didelphis marsupialis*) | 19 | 1 | 1 |
| Procyonidae | Common racoon (*Procyon lotor*) | 1 | | |
| Canidae | Gray fox (*Urocyon cinereoargenteus*) | 8 | 6 | 6 |
| Mephitidae | Hooded skunk (*Mephitis macroura*) | 1 | | |
| Felidae | Jaguarundi (*Puma yagouaroundi*) | 1 | | |
| Momotidae | Lesson's motmot (*Momotus lessonii*) | 1 | | |
| Dasypodidae | Nine-banded armadillo (*Dasypus novemcinctus*) | 12 | | |
| Felidae | Ocelot (*Leopardus pardalis*) | 8 | | |
| Mustelidae | Tayra (*Eira barbara*) | 2 | | |
| Procyonidae | White-nosed coati (*Nasua narica*) | 1 | | |
| | Total | 54 | 7 | 7 |

**North Carolina, USA**

| Family | Common Name (Scientific Name) | Encounters | Detections | Attacks |
|---|---|---|---|---|
| Corvidae | American Crow (*Corvus brachyrhynchos*) | 2 | | |
| Ursidae | Black bear (*Ursus americanus*) | 19 | 7 | 5 |
| Procyonidae | Common racoon (*Procyon lotor*) | 80 | 17 | 4 |
| Canidae | Gray fox (*Urocyon cinereoargenteus*) | 29 | 5 | 2 |
| Didelphidae | Virginia opossum (*Didelphis virginiana*) | 15 | 3 | 1 |
| Phasianidae | Wild Turkey (*Meleagris gallopavo*) | 2 | | |
| | Total | 147 | 32 | 12 |

Note:
Frequency of encounters, detections, and attacks by each snake predator species observed from camera trap videos during three field experiments conducted in Ecuador, Mexico, and North Carolina, USA, that were aimed to test hypotheses of aposematism and mimicry. Nomenclature follows *Ridgely & Greenfield (2001)*, *Wilson & Reeder (2005)*, *Peterson (2010)*, and *Vallely & Dyer (2018)*.

North Carolina, activity peaked at night, with most attacks occurring just after sunset (Fig. 5). In Mexico, predator encounters were more common at night; however, most detections and attacks occurred during the day (Fig. 5).

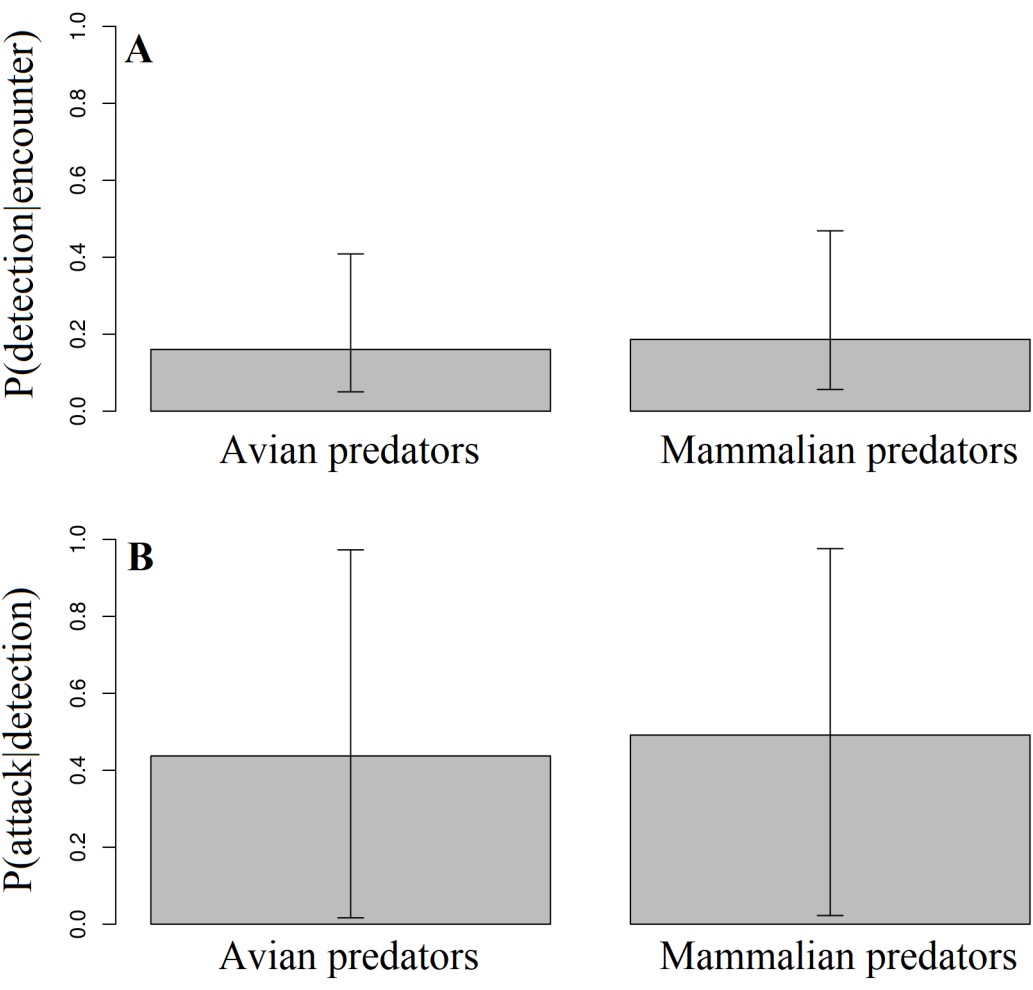

**Figure 4 Detection and attack probabilities of avian vs. mammalian predators.** The probability that avian vs. mammalian predators detected replicas after encounter (A) and attacked replicas after detection (B) across all experimental locations.

## Variation among predator species

The frequency and timing of encounters, detections, and attacks also varied among predator species. In Ecuador, activity was dominated by collared peccaries (*Pecari tajacu*), white-lipped peccaries (*T. pecari*), and gray-winged trumpeters (*Psophia crepitans*) (80.5% of encounters, 88.9% of detections, and 100% of attacks; Table 2). In Mexico, activity was dominated by common opossums (*Didelphis marsupialis*), gray foxes (*Urocyon cinereoargenteus*), and nine-banded armadillos (*Dasypus novemcinctus*) (72.2% of encounters, 100% of detections and attacks; Table 2). In North Carolina, activity was mostly restricted to black bears (*Ursus americanus*), common racoons (*Procyon lotor*), Virginia opossums (*Didelphis virginiana*), and gray foxes (97.3% of encounters, 100% of detections and attacks; Table 2).

A total of 11 of 20 predator species (five bird species and six mammal species) that were encountered never detected replicas (Table 2). Each of these species was encountered 10 times or less (mean ± s.d.: 2.27 ± 2.72; median = 2; Table 2). In contrast, nearly all of the

**Table 3 Camera trap observations.**

|  | Ecuador | Mexico | North Carolina | Total |
|---|---|---|---|---|
|  | 402 | 630 | 504 | 1,536 |
| Predator encounters | 16.7 (67) | 8.6 (54) | 29.2 (147) | 17.4 (268) |
| Mammalian predator encounters | 11.7 (47) | 8.4 (53) | 28.4 (143) | 15.8 (243) |
| Avian predator encounters | 5.0 (20) | 0.2 (1) | 0.8 (4) | 1.6 (25) |
| Detections | 4.0 (16) | 1.1 (7) | 6.3 (32) | 3.4 (52) |
| Mammalian predator detections | 3.0 (12) | 1.1 (7) | 6.3 (32) | 3.1 (48) |
| Avian predator detections | 1.0 (4) |  |  | 0.3 (4) |
| Attacks | 0.5 (2) | 1.1 (7) | 2.4 (12) | 1.4 (21) |
| Mammalian attacks | 0.2 (1) | 1.1 (7) | 2.4 (12) | 1.3 (20) |
| Avian attacks | 0.2 (1) |  |  | 0.1 (1) |
| Attacks recorded on clay but not cameras | 0.2 (1) |  | 0.99 (5) | 0.39 (6) |
| Attacks recorded on cameras but not clay | 0.5 (2) | 0.63 (4) | 0.4 (2) | 0.52 (8) |
| Attacks recorded on both cameras and clay |  | 0.48 (3) | 1.98 (10) | 0.78 (12) |
| Number of replicas with functional cameras | 34 | 21 | 54 | 109 |
| Number of undetected replicas | 24 | 14 | 29 | 67 |
| Number of marks on replicas with cameras |  | 3 | 15 | 18 |

Note:
Frequency of encounters, detections, and attacks are in behavioral events/100 camera days (total number of observations is given in parentheses). Number of camera days is given below the site headings. Numbers of encounters, detections, and attacks are based on records separated by at least 30 min (for a given species at a given site).

nine species of predator (one bird species and eight mammal species) that detected replicas were commonly encountered near replicas (mean ± s.d.: 26.11 ± 22.40; median = 19; Table 2). Species with the highest detection per encounter rates were *Pecari tajacu* (42.3%), *Ursus americanus* (36.8%), and *Urocyon cinereoargenteus* (29.7%) (Table 2). Species with the lowest detection per encounter rates included ocelots (*Leopardus pardalis*; 0.0%), *Didelphis marsupialis* (5.2%), and *Dasypus novemcinctus* (11.1%) (Table 2). Of species that detected replicas at least five times, the highest attack per detection rates were by *Urocyon cinereoargenteus* (72.3%) and *Ursus americanus* (71.4%) (Table 2). Species with the lowest attack per detection rates were *Pecari tajacu* (0.0%) and *Procyon lotor* (23.5%) (Table 2).

## Clay marks vs. videos

Using marks left in clay replicas, we observed 33 avian attacks and 21 mammal attacks in Ecuador, 78 avian attacks and 92 mammal attacks in Mexico, and 16 avian attacks and 198 mammal attacks in North Carolina (Fig. 6). A total of 18, 57, and 12 replicas from Ecuador, Mexico, and North Carolina, respectively, were scored as missing, as we were not able to locate any trace of these replicas at their original location (Fig. 6). Using video, we observed one avian and one mammal attack in Ecuador, seven mammal attacks in Mexico, and 12 mammal attacks in North Carolina (Fig. 7; Table 3). We found that marks left in clay replicas revealed a significantly higher ratio of avian: mammalian attacks than camera trap videos (Fisher's exact test; $p = 0.012$).

Across all experimental locations, 13 replicas that were registered as attacked based on videos were also scored as attacked based on clay marks (Table 3). Eight replicas that were

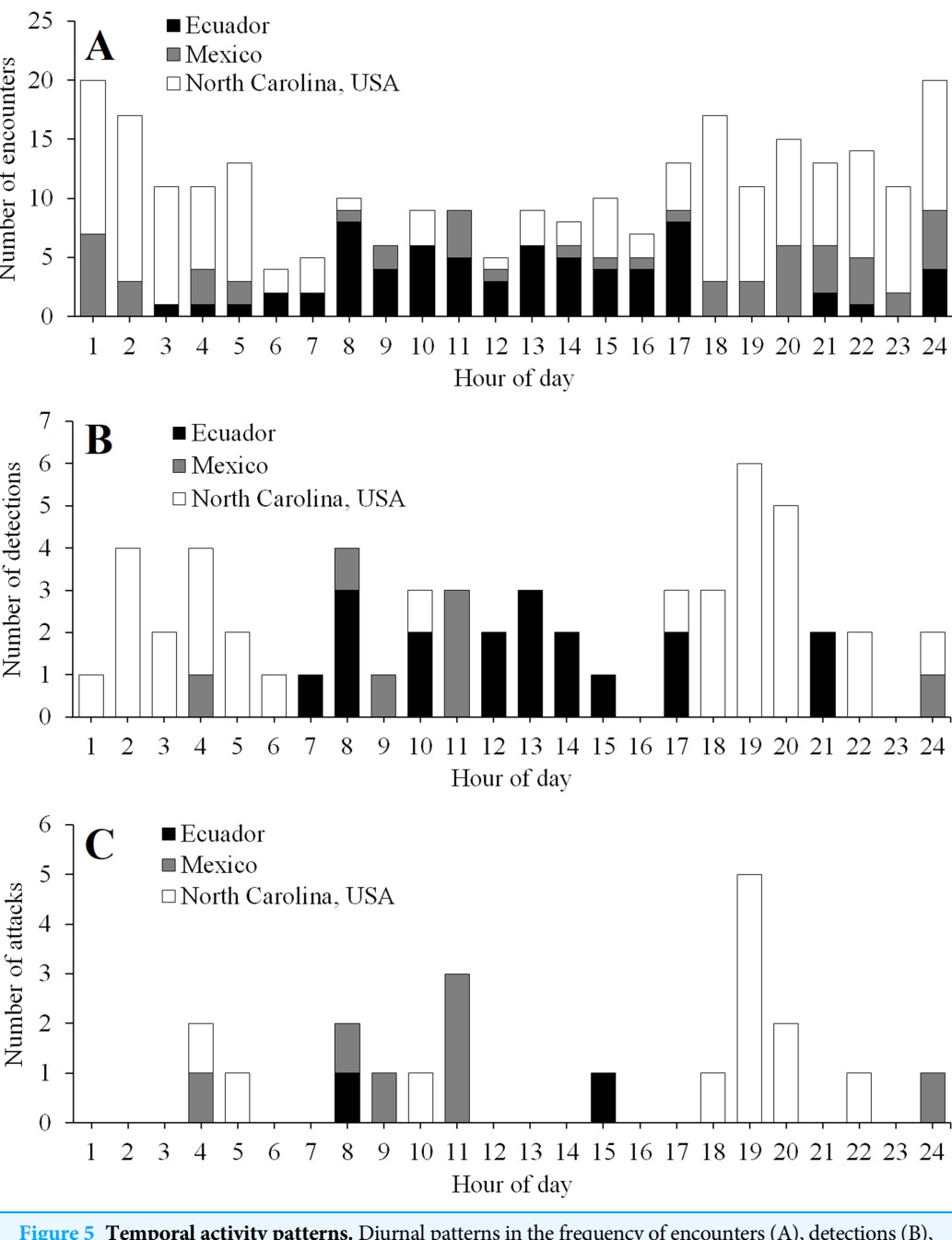

**Figure 5 Temporal activity patterns.** Diurnal patterns in the frequency of encounters (A), detections (B), and attacks (C) in field experiments conducted in Ecuador, Mexico, and North Carolina, USA. Daytime ran from 6 to 18, 6 to 19, and 8 to 17 h in Ecuador, Mexico, and North Carolina, USA, respectively.

registered as attacked based on videos were not scored as attacked using clay marks (Table 3). In five of these cases, replicas were scored as missing in the field as videos confirmed that predators removed replicas from their original location. In two cases, replicas were present but no impressions indicative of bite marks were visible. In a final case, one predator attacked a replica without destroying it and another predator later attacked the same replica; thus, this replica was scored as having two attacks according to

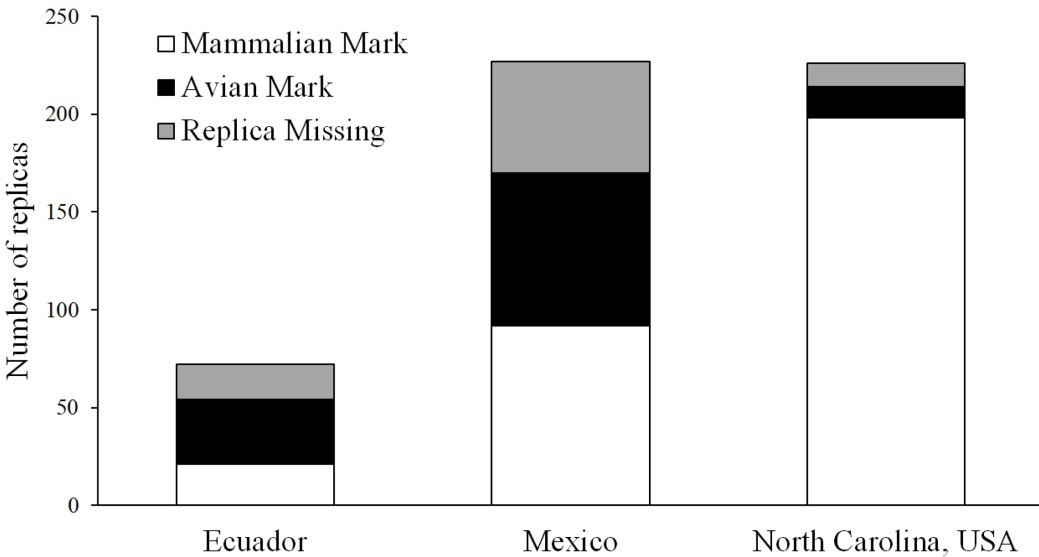

**Figure 6 Results of field experiments.** Numbers of replicas—both with and without camera traps—that bore marks indicative of attacks by avian and mammalian predators as well as numbers of replicas that were missing (i.e., no trace of replica found) in field experiments conducted in Ecuador, Mexico, and North Carolina, USA.

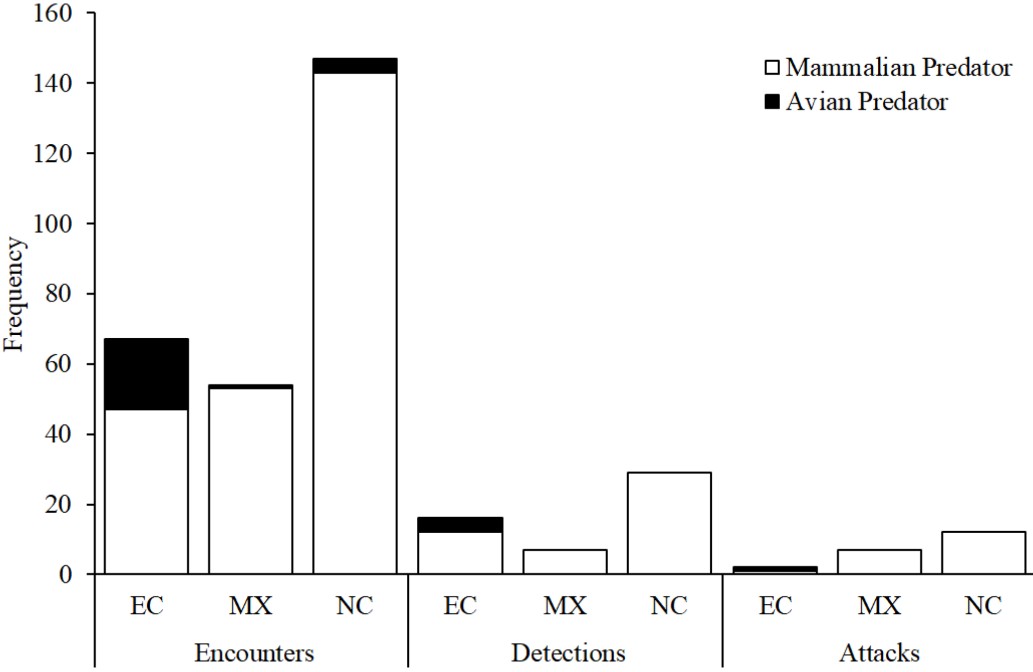

**Figure 7 Camera trap observations.** Numbers of encounters, detections, and attacks by avian and mammalian snake predators observed from camera trap videos at each experimental location: EC, Ecuador; MX, Mexico; NC, North Carolina, USA.

video but only one attack was scored based on clay marks. No evidence of attacks by predators was obtained from videos for six replicas that were scored as attacked based on clay marks (Table 3).

## DISCUSSION

We evaluated whether camera trap videos can provide additional information that could be useful to field studies of predation employing artificial prey. Field studies typically rely on the relative frequencies of clay marks on different prey phenotypes to infer avoidance behaviors of predators (*Noonan & Comeault, 2008*; *Marek et al., 2011*; *Dell'Aglio, Stevens & Jiggins, 2016*; *Kristiansen et al., 2018*). Previous predation field studies that have employed camera traps have generally used photography (Fig. 1), have been conducted on small scales, and have primarily employed cameras for the sole purpose of identifying predators attacking artificial prey. Our observational data collected from three field experiments conducted in three separate locations show that camera trap videos can be used to provide benefits to field studies of predation beyond predator identification.

Our study demonstrates how data on the frequency at which different predator species encounter, detect, and attack replicas could be gathered using videography. These data could be used in a variety of ways to enhance predation studies employing artificial prey.

First, these observational data could be used to make more robust evaluations of the relative fitness of different prey phenotypes. For example, in heavily shaded habitats such as the tropical forests where field experiments were conducted in Ecuador and Mexico, the warning coloration of coral snakes and their mimics is unlikely to provide protection from predation at night given that the visibility of their color patterns to predators should be low (*Kelber, Yovanovich & Olsson, 2017*). Information on warning coloration is therefore unlikely to factor into decisions by predators to attack replicas at night in such habitats. As a result, an analysis that omitted the two attacks that were observed at night in Mexico (Fig. 5C) would provide a more robust test of how warning coloration factors into prey-selection decisions by predators. Similarly, because different color pattern phenotypes might vary in their conspicuousness to predators, differences in predation rates could be driven by both variation in prey preference and variation in visual detection rate (*Stuart, Dappen & Losin, 2012*; *Rojas, Rautiala & Mappes, 2014*). Variation in visual detection rate has been shown to be an unlikely explanation for differences in predation rates between color pattern phenotypes in at least a few aposematic taxa (*Brodie, 1993*; *Wüster et al., 2004*; *Buasso, Leynaud & Cruz, 2006*; *McElroy, 2016*). Nevertheless, restricting analyses to replicas that were actually detected would provide more direct tests of the fitness consequences associated with different prey phenotypes, given that the fitness benefits of aposematic prey should only be realized after predators have detected prey. Replicas monitored by cameras across all field experiments more often remain undetected than detected throughout the monitoring period (Table 3). Thus, field studies of aposematic prey that limited analyses to the subset of detected replicas could potentially benefit from increased statistical power to resolve differences in predation between phenotypes.

Second, these observational data could be used to more precisely characterize how different predators contribute to selection on prey phenotypes. Although predator communities as a whole did not have a tendency to attack or avoid replicas following

detection (Fig. 4), the data tentatively suggest that predators might vary in their behavioral responses to aposematic phenotypes (Table 2). At least one predator species, *Peccari tajacu*, had a tendency to disproportionately avoid coral snake phenotypes, while most other predator species (e.g., *Urocyon cinereoargenteus*) attacked them (Table 2). Given that *Peccari tajacu* is largely diurnal and is one of the most common predators at Tiputini Biodiversity Station in Ecuador (*Blake et al., 2012*; *Blake & Loiselle, 2018*), their contribution to selection might be disproportionately small relative to their abundance. Likewise, *Urocyon cinereoargenteus* is one of the more common mammals encountered during camera trap surveys conducted in the Carolina Sandhills (C. Akcali and D. Pfennig, 2015, unpublished data), where they are largely crepuscular and nocturnal like the coral snake mimics with which they co-occur (*Palmer & Braswell, 1995*; *Whitaker, 1998*). Consequently, *Urocyon cinereoargenteus* might have been a key predator in facilitating the recent rapid evolution of a coral snake mimic in the Carolina Sandhills (*Akcali & Pfennig, 2014*). However, these claims remain speculative until additional data are gathered that permit a more robust characterization of the prey selection functions of these predators.

Third, observational data from videos could allow more data to be collected from artificial prey experiments. When no traces of a replica can be located at their original location, researchers often conservatively score such replicas as missing and omit them from subsequent analyses (*Kikuchi & Pfennig, 2010*; *Chouteau & Angers, 2011*; *Lawrence, Mahony & Noonan, 2018*). However, videography—more often than photography—can provide conclusive evidence of cases where missing replicas were due to removal by predators. Across all three experiments, videos revealed that all six replicas that were scored as missing in the field were actually removed by predators. Given that a total of 87 replicas were scored as missing across all three field experiments (Fig. 6), the potential for videos to rescue lost data might be substantial.

Fourth, these observational data could provide insight into the extent to which artificial prey approaches sample a biased subset of the predator community. Several studies have suggested that avian predators should be more important selective agents on coral snake color patterns than mammalian predators, especially in the tropics (*Brodie, 1993*; *Brodie & Janzen, 1995*; *Hinman et al., 1997*). During our field experiments, avian predators were substantially underrepresented on videos relative to the frequency at which their beak marks were recorded on replicas that were not monitored by cameras (Figs. 6 and 7). This pattern is generally consistent with most camera trapping studies that report capture rates for both mammalian and avian species, which have found that avian species tend to have lower capture rates on cameras (*Stein, Fuller & Marker, 2008*; *Blake et al., 2011*; *Naing et al., 2015*). Thus, it is not clear whether this difference in the representation of avian predators in videos and clay marks reflects the fact that avian predators often moved too fast to be recorded on videos, that avian predators detected replicas outside the field of view of the cameras and actively avoided cameras as a consequence, or alternatively, that this was simply due to the low number of cameras relative to replicas that were not monitored by cameras (Table 1). Avian predators and some mammalian predator species (e.g., *L. pardalis*; Table 2) might have extremely

low rates of detections relative to encounters. Predators with low detection rates might not be capable of being sampled using artificial prey approaches either because replicas do not provide the cues needed for predators to easily detect them or because these predators detect replicas but do not classify them as edible prey. In such cases, laboratory experiments might be necessary to definitely characterize the ability of predators to detect replicas (*Rößler, Pröhl & Lötters, 2018*). Predator species that are infrequently captured on video would be particularly important for controlled experiments given that low encounter rates ultimately preclude assessment of predator sampling biases of artificial prey.

Thus, videography can provide some additional information for artificial prey studies, but is it worth the costs? An informal survey of predation studies employing artificial prey (see Fig. 1 for search details) revealed that—out of studies that report both sample sizes and the length of time artificial prey were exposed to natural predators ($N = 441$ studies)—most employ large numbers of replicas (mean ± s.d. = 482 ± 712, median = 300) for an exposure period close to 2 weeks (mean ± s.d. = 12.7 ± 9 days, median = 12 days). Although the amount of information provided by videos varied substantially among our experiments (Fig. 7; Table 3), one camera, averaged across all three experiments, obtained 0.18 encounters, 0.04 detections, and 0.01 attacks per day by species that we classified as predators. If these frequencies are calculated over a single transect consisting of 30 video-monitored replicas, which would represent 10% of the total replicas employed in the median artificial prey experiment, over a 12-day study timeline, representing the length of the median artificial prey experiment, a total of 65.3 encounters, 13.7 detections, and 4.8 attacks would be expected to be observed. If each camera were to cost $100, each additional encounter, detection, and attack in terms of camera expenses would cost approximately USD $46, $219, and $625, respectively. If these figures were to be calculated for avian predators alone, a total of 7.1 encounters, 1.2 detections, and 0.3 attacks would be expected for a single 30-replica transect monitored by cameras for 12 days, with each additional encounter, detection, and attack requiring USD $423, $2,500, and $10,000, respectively, in camera costs. Thus, obtaining additional information via videography can be relatively expensive even without considering its accompanying logistical and time costs, which are not negligible but relatively minor comparatively speaking (Table S2). Indeed, the cost of cameras that was incurred for each of our field experiments was more than the total cost of conducting any one experiment without cameras (Table S2). The reliability of video recording can impose additional costs, as six out of 18 replicas monitored by cameras bore clay marks by predators but no evidence of predation was captured on video.

In other systems, however, these costs might not be quite as high. If the percent of replicas attacked per day is used as a proxy for predator activity, the average predator activity level from our three experiments (ca. 1% replicas/day) was lower compared to other artificial prey studies (mean = 6% replicas/day, median = 4% replicas/day, $N = 424$ studies). If we recalculate the amount of information and costs that would be expected for a single transect of the median artificial prey study (30 camera-monitored replicas for 12 days) assuming that differences in encounters, detections, and attacks are proportional

to differences in encounters, detections, and attacks that were estimated in our study, a total of 98 encounters, 20.6 detections, and 7.2 attacks would be expected, with each additional encounter, detection, and attack requiring approximately USD $31, $146, $416, respectively, in camera costs. If these same calculations and assumptions are made using each of the predation rates that have been reported from our informal literature survey, the minimum level of predator activity (in terms of % predation per day) necessary for the purchase of one additional camera to capture an additional encounter, detection, or attack would be approximately 0.01%, 0.03%, and 0.08% replicas/day, respectively (Fig. S2). Approximately 68.3% of artificial prey studies have reported predator activity levels higher than the 0.03% threshold, whereas only 18.2% of such studies have reported predator activity levels higher than the 0.08% threshold. Unless measures are taken to increase the rate at which information could be obtained (e.g., increasing the realism of replicas; *Paluh, Hantak & Saporito, 2014*), the benefits of additional information would only be worth the cost of cameras in a minority of systems.

## CONCLUSIONS

Results from this study provide quantitative estimates of the amount of additional information that camera trap videos could provide to artificial prey studies and demonstrates some of the benefits of using videography over remote photography in artificial prey studies. Across three field experiments, dozens of observations were obtained on the frequency at which predators encounter, detect, attack, and avoid artificial prey. Observations of predator activity were dominated by mammals. Videography likely underestimates activity by avian predators as marks on artificial prey registered a higher ratio of avian:mammalian attacks than videos. These observational data can be used to estimate the rates and probabilities of encounters, detections, attacks, and avoidances by predators. This information could then be used to conduct more direct tests of the relative fitness of different artificial prey phenotypes as well as provide insight into the relative contribution of different predator species to selection on prey. However, the incorporation of cameras into artificial prey studies that experience low rates of predator activity would be difficult to justify given the current costs of cameras. Nevertheless, videography would still prove useful at smaller scales as a tool to generate new observations that could lead to new questions or ideas for testing.

## ACKNOWLEDGEMENTS

We thank D. Kramer for providing comments that significantly improved the manuscript. We also thank J. Hunter, C. Porter, and one anonymous reviewer for comments. We thank B. Edwin Siurob-Espíndola, C. Iván Hernández-Herrera, D. Joaquin Sánchez-Ochoa, E. Hernández-Martínez, S. Jovita González-Ramos, J. Vaca Guerrero, and personnel at Tiputini Biodiversity station for help with fieldwork. We thank the Ecuadorian and Mexican government for providing the necessary research permits.

### Funding

This work was supported by a Student Research Grant from the Animal Behavior Society, the National Science Foundation (No. 1643239), and Reynolds Competitive Research Leave, College of Arts and Sciences, University of North Carolina at Chapel Hill. The funders had no role in study design, data collection and analysis, decision to publish, or preparation of the manuscript.

### Grant Disclosures

The following grant information was disclosed by the authors:
Animal Behavior Society.
National Science Foundation (No. 1643239).
Reynolds Competitive Research Leave, College of Arts and Sciences, University of North Carolina at Chapel Hill.

### Competing Interests

The authors declare that they have no competing interests.

### Author Contributions

- Christopher K. Akcali conceived and designed the experiments, performed the experiments, analyzed the data, contributed reagents/materials/analysis tools, prepared figures and/or tables, authored or reviewed drafts of the paper, approved the final draft.
- Hibraim Adán Pérez-Mendoza performed the experiments, contributed reagents/materials/analysis tools, approved the final draft.
- David Salazar-Valenzuela performed the experiments, contributed reagents/materials/analysis tools, approved the final draft.
- David W. Kikuchi analyzed the data, approved the final draft.
- Juan M. Guayasamin contributed reagents/materials/analysis tools, approved the final draft.
- David W. Pfennig contributed reagents/materials/analysis tools, approved the final draft.

### Field Study Permissions

The following information was supplied relating to field study approvals (i.e., approving body and any reference numbers):

Fieldwork was done under the following permits: Ecuador–N° 002-017 IC-FAU-DNB/MA; Mexico–SGPAJDGVS/09347/16. No permits were required in North Carolina, USA.

### Data Availability

Akcali C. 2019. Literature_Survey_Data_12.16.18.xlsx. DOI: 10.6084/m9.figshare.7647131.v1
Akcali C. 2019. Clay_Mark_Data_12.16.18.xlsx. DOI: 10.6084/m9.figshare.7647128.v1.

Akcali C. 2019. Camera_Trap_Scores_12.16.18.xlsx. DOI: 10.6084/m9.figshare.7647113.v1.
Raw data is also available included in the Supplemental Information.

## Supplemental Information

Supplemental information for this article can be found online at http://dx.doi.org/10.7717/
peerj.6487#supplemental-information.

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
