# Peer review of "Evaluating the utility of camera traps in field studies of predation"

_PeerJ, doi:10.7717/peerj.6487_

## Round 0.1 · original submission · Major Revisions

Overview
This study investigated the extent to which video recording provided additional information about the interactions between potential predators and artificial snake prey (replicas). The data were collected from video-recording camera traps in three locations during studies of aposematism and mimicry in coral snakes and their mimics. During 1,022 days of recording (Ecuador 392, Mexico 126, USA 504) [number of recording days not completely clear in manuscript], 11 attacks on replicas were indicated by marks in the clay (Ecuador 0, Mexico 1, USA 10), and 5 replicas went missing (1, 4, 0), presumably removed by predators. In contrast, the videos recorded 257 potential predators in the vicinity of the models (55, 54, 148) and 18 attacks (2, 6, 10) as well as 34 avoidances (12, 1, 21). In 8 cases (0, 1, 7), replicas were removed without any video record. Although evidence from the larger set of replicas, including those without video recording, indicated that only about half of the attacks at two sites were generated by mammals with the rest from birds, the great majority of potential predators recorded near and interacting with the replicas were mammals. At the third site, mammals were the majority of those leaving marks on the replicas as well as present in the area and interacting with the replicas. In addition to more precise identification of attacking predators and of the number and species of animals that pass near the replicas without interacting with it, video allowed discrimination between animals that detected but did not attack a replica and those that did not detect a replica. This can provide important insights into the effect of morphological characteristics and microhabitat of prey on interactions with their predators. However, it is possible that video cameras in some way bias the relative number of birds and mammals that attack the replicas.

The reviewers and I consider that the manuscript has the potential to make a sound contribution to studies of predation using artificial prey. However, we feel that statistical support for observed patterns, greater attention to terminology and possibly a substantial reorganization would make its message clearer and more decisive. I have provided detailed comments below. You may take these as if I were a third reviewer: make appropriate changes if I have made a valid point and explain why you did not make changes if my comments are not valid. In addition, I have attached a pdf with comments on word use, grammar and more concise writing. You do not have to respond to these suggestions in detail unless you choose not to make the changes.

Major Concerns

1) Use of the term ‘experiment’. The data presented here are not experimental in the true sense of the term. Because they do not involve manipulation of a proposed causal variable along with a control, they are observational data. I don’t doubt that your studies of aposematism and mimicry were true experiments but simply that the present manuscript is not the result of experimental data. I suggest that when you introduce the present study at the end of the Introduction, you indicate more clearly the purpose of the three experiments that were the source of the data (the text is much more appropriate than a table) and that you extracted data from those studies for the present investigation. The term ‘experiment’ is acceptable when referring to the original studies but not when referring to the observational data (e.g., L248; your study is not providing experimental support for this conclusion).
2) Explaining the study system. I agree with Rev 1 that the manuscript would benefit from a paragraph that provides the context of your study. Not all readers will recognize that coral snakes are venomous and that their coloration may be a warning to potential predators. This would also allow you to emphasize the particular importance of avoidance following detection in your system. This paragraph would be strengthened by an illustration in supplementary material of a real snake and one of your models. However, I don’t think you have to go so far as to add illustrations of all species; if you provide the names, interested readers can find photos, and the appearance of the snakes and replicas is not considered in this manuscript and should be reserved for other publications on the results of your experiments. With regard to potential predators, I accept that there will not be published accounts of all potential predators. A short statement on the range of species that may be expected to prey on snakes of this size range might be useful, including a note with references to indicate that this list might include species not typically thought of as predators. If there is evidence from other studies that replicas may be attacked by animals that are not considered predators of that prey type, it might be useful to mention that. Once that is clarified, I have no objection to continued use of the terms ‘predator’ and ‘prey’, although ‘potential predator’ might be more appropriate. While the reviewer’s comments would be highly relevant to your publications on the effect of color on predator responses, your focus here is on the ability to identify attacks and avoidances, so whether or not they are true predators is not critical.
3) Sample size. I found it difficult to fully understand the sample size (and indeed may have erred in my overview summary). For example, I wanted to know the sampling effort, Table 1 indicated that in Ecuador there were 32 camera traps operating for 14 d (448 camera-days). L184 indicated that 4 cameras failed (448-56=392 camera-days), but Table 2 indicates 414 camera-days. It is important that sample sizes and how they were calculated for each step in the study, i.e., trap-days, encounters, detections, attacks and avoidances, be clear. It is possible that a more logical organization of Table 2 would make this more transparent.
4) Terminology. It is also important that you select your terms carefully and be sure that the reader understands them.
• For example, detection is defined in terms of a potential predator’s behavior, but Table 2 repeatedly refers to detection in the context of the researcher detected a potential predator. This can be very confusing.
• On L175, the reference to visitation suggests that you mean researcher detection of animal encounters, which is really encounter rate, but you could also mean rate of detections of replicas by animals.
• ‘Video’ is such a general term in this study, that ‘number of videos’ does not have an unambiguous meaning, despite an attempt to clarify on L173.
• Furthermore, because you use data from all replicas to examine the relative importance of predation attempts by mammals and birds, the reader needs to know unambiguously the sample size of replicas. It is not clear if the total number given in Table 1 simply needs to be multiplied by number of days to get a replica-day measure equivalent to camera-days. I suspect it doesn’t because attacked replicas had to be replaced. If so, we should have the number of sites with replicas that were not video recorded and the number of days they were left and the resulting number of replica-days, with perhaps a reminder that missing/damaged replicas were not replaced and perhaps an estimate of the effective number. We should know the number of these that were attacked and missing and by which category of predator. Since you will be publishing the main results separately, I presume, you have to be careful to avoid the appearance of double publishing. However, I don’t think this basic summary data will threaten your main study, especially if you are summarizing across all replica types.
5) Study questions and statistical support
• I think that your study could be much more tightly organized around a logical series of questions and that some statistical comparisons would be appropriate, as suggested by the reviewers. Most of the issues mentioned below were referred to or implied in your manuscript but never with statistical support and often without even adequate descriptive documentation to allow a reader to fully understand your reasoning.
• Using the term ‘encounter’ as in optimal foraging models (coming close enough to potentially detect a prey, whether or not the prey is detected), would allow you to ask how many encounters occurred per camera-day, how many detections occurred per encounter, how many attacks and avoidances occurred per detection. Detections and avoidances are particularly important because these are the two variables that are unavailable with replicas alone or replicas with still cameras. It would be great if you could determine some way to provide confidence intervals for these numbers.
• With regard to species, you could ask what was the total number of individuals of each species that encountered, detected, attacked and avoided replicas and indicate whether there were changes in distribution of species among these stages, potentially supported by statistical analysis. (You implied that this was the case but did not provide detailed documentation such as the number of ocelot encounters and detections.)
• You could ask whether the total number of mammal and birds attacks differed from the totals for encounter and detections and from the distributions indicated by marks on the replicas with cameras and on the non-camera replicas. I think that this could be tested statistically. Given the relative lack of birds in the video recordings, it is important to know if this is reflected in the damage to recorded replicas as opposed to unrecorded.
• It would seem possible that you could provide statistical support for the statement that avoidance was more frequent than attack for detected prey, but I am not certain that statistical support is needed for this comparison. If you could calculate confidence intervals for your means, that might be sufficient.
• I do not agree with the reviewer that you could compare locations. Because the study was not designed as a regional comparison and differed in design between regions, I do not think that this would be informative.
• I also do not agree with the reviewer that you examine issues such as time to detection in relation to predator density. I think that time to detection is not necessarily a particular benefit to video analysis and is beyond the scope of the present manuscript.
• Finally, you might want to include a question about the reliability of video recording (i.e. frequency with which interactions were missed by video). This could be assessed by presence of marks on replicas that had camera traps but for which there was no recorded interaction. Even if not presented as a research question, this is an important issue that should be included in the Discussion.
6) Organization of the Results. At present, it is very hard to quickly and clearly grasp your findings. I believe that this is, at least in part, because the main message of your paper concerns the ability to document encounter, detection, attack, and avoidance patterns in relation to the replicas whereas the manuscript results are organized according to study locations. Papers generally read better when the questions raised in the Introduction provide the organizational structure for subsequent Methods, Results and Discussion. Therefore, I suggest giving careful consideration to a major reorganization, perhaps along the lines of the outline in my overview. However, if you do not undertake a revision of the organization, it is very important that you strengthen the organization and clarity of the data presentation for each location to make sure that readers absolutely understand what is being presented and can easily find the equivalent information in each sub-section. This will be aided by very explicit questions, clear definitions and consistent order of information.
7) Discussion of benefits and costs of additional camera traps. The scaling up of video on L252-258 early in the Discussion is too long, out of place, and probably not needed at all. The potential number of attacks that could be observed is an obvious product of attack rate and number of locations. This concern also applies to L261-263 and L294-322. I suggest that after discussing each of the components of behavior and being clear that the reader understands the rate of each in each location, you devote a single paragraph to the benefits and costs of adding video cameras to sites. This might involve an estimated number of events of each type recorded per additional camera (or some larger number of cameras if it results in fewer decimal places) and the equivalent financial and time costs. Note that providing rates per recorded site will be much more useful than the potential absolute number you would have found with your particular study design.

Specific Comments

Key Words
I suggest adding the families of the snakes and the names of the three study locations.

Abstract
L28. I disagree with Rev 1 here; using the term ‘artificial prey’ in a general statement seems appropriate.
L30. I agree that ‘unmarked prey’ should be replaced by a clearer term.
L30. You can make a stronger statement than ‘interactions between predators and prey’ here. I suggest indicating presence of potential predators near the artificial prey and their identification, evidence for detections that resulted in no attack, and improved identification by potential predators that did attack the prey. But some of these advantages also occur with still cameras and thus are not unique to video.
L35-36. Here, I think you can briefly indicate the general type of prey (coral snakes and mimics, including scientific names of genera), habitat types, and locations (name of park and country only) which would add important details of your study and not add many more words.
L35. Clarify that during three experiments on aposematism and mimicry you used video camera traps near some of the artificial prey to examine how much additional information videos would provide. The word ‘site’ could refer to study site (e.g., a reserve in Ecuador), so you need less ambiguous terminology.
L36-39. If you follow advice from me and the reviewer regarding statistical analysis and clarification of the research questions, the Abstract will become more specific here.
L39-40. The reviewer is correct that ‘studies of predation’ is too broad. Try to come up with a more focused statement on your contribution to studies of artificial prey.

Introduction
L53. The explanation on L60-65 goes more logically here before the variety of research questions and taxa.
L69. Not clear how a footprint is a disturbance. Do you mean actual marks on a clay model from it being stepped on? Is that very likely?
L73. Since your approach allows specific identification of potential predators, shouldn’t you emphasize that marks on clay do not typically allow species identification but only broader groups of potential predators?
L94-101. This section synthesizing previous work needs to be strengthened. Although you refer to the existence of several previous studies, you cite only one. It would be appropriate to cite the major recent studies and perhaps the first one. Furthermore, by saying that most used still images, the reader is lead to believe that there exist previous but uncited studies using videography. Such studies should be cited. Then, you should develop the knowledge gap to be filled by your study. Is this the first systematic attempt to quantify the benefits of videography or a repetition of previous studies in different habitats or artificial prey types or with a larger sample size?
L99-101. This is a conclusion of your study but presented here as if already known. It seems logical that it would be superior, but then what is the point of your study? If you identified some potential difficulties of video, it would make sense not just to find out if the technique was an improvement but how much of an improvement and what limitations there might be. If little is known about optimal arrangements of camera and prey for maximizing information gain, they could also be a contribution.
L102-104. This would be a good place for a brief overview of the primary focus of the three studies and the use of the data to ask one or more different questions.
L104-109. Although Rev 1 suggests removing these comments as Methods, I believe that a few overview sentences can help readers understand the more detailed methods to follow, so you can leave them in if you wish. Rev 1 also asks you to list any hypotheses you had. Given the descriptive nature of this study, I can see that you might not have had hypotheses, but simply decided after the study that it would be good to quantify the gains from video. Only state questions asked as hypotheses if they were truly developed before the studies were carried out. If you follow my advice to focus your study questions more precisely, you can be more specific in the questions asked here.

Methods
L115. Why is there no permit for the US study?
L119. I agree with Rev 1 that you should provide more somewhat more detail about the goal of each experiment, the common and scientific names of the species involved, and the sizes of the replicas.
L129-130. Provide full names of the museums, here or in supplemental material.
L133. I suggest removing from Table 1 location details (leaving just the country for reference), latitude and longitude, dates, habitat types, elevation, study species and subject of study as these either already overlap with information in the text or that can be more clearly presented in the text. Add a sentence at the end of the paragraph (L143) citing Table 1 for additional details. Consider whether the order of all the rows is logical for a reader. For example, it seems as though the sequence number of transects, distance between transects, and placement of replicas within transects would be easier to follow. As details of cameras are in the text, you only need to indicate the number and make. I suggest the information about clustered vs isolated replicas and camera distance would be better as text as you make a point about its impact later. Finally, most journals, including PeerJ, do not use vertical lines in tables. Please remove these lines and develop a format that allows the information to be clear without them. In general, horizontal lines are not required either unless the table needs to be subdivided into larger groups.
L143. Add a new paragraph here to clearly explain your sample size. Number of cameras, number of recording failures, number of days to come up with a total of camera days for each location.
L144-169. I proposed some changes to the text to make it more concise and clear, and to avoid the implication that you could assess an animals motivation as opposed to its behavior. Don’t use these changes unless I correctly captured your meaning. If I failed to do so, my suggestions should still reveal where there seemed to be a problem.
L157ff. In the section describing your categorization of behavior, you should cite the video clips in your supplementary material. Furthermore, I think it would be extremely useful not to restrict the videos to attacks but to also include one or more examples of detection followed by avoidance.
L166,168. I suspect that the term ‘footage’ is slang and/or out-of-date (referring to movie film). Check on the correct term for a video recording.
L169. Note that you refer to interaction here but have not explicitly defined it above. Please add the operational definition, perhaps at the start of the paragraph on L157.

Results
L173. Given that you are providing video clips in the supplementary material, I question whether these photos add anything substantial to the manuscript.
L180. I believe that you mean species richness rather than diversity.
L186. What was the frequency of mammal and bird marks on the replicas with cameras?

Discussion
L252-258. The scaling up of video is out of place here. The potential number of attacks is an obvious product of attack rate and number of locations. I suggest that after discussing each of the components of the interaction process and being clear that the reader understands the rate of each in each location, you devote a single paragraph to the benefits and costs of adding video cameras to sites. This might involve an estimated number of events of each type recorded per additional camera (or some larger number of cameras if it results in fewer decimal places) and the equivalent financial and time costs. Note that providing rates per camera will be much more useful than the potential absolute number you would have found with your particular study design. It would be useful to be aware of the precision (or lack of it) in these rates when making your estimates.
L267, 269. The problem with lumping failure to detect and avoidance is not really a statistical issue but a conceptual one that could lead to incorrect inferences.
L271ff. This paragraph needs development. You are referring to a bias without a clear indication that there is one. The bias to which you are referring is ambiguous. Do you mean that video recording may be biased against birds by failing to record their presence when they are actually there or that they avoid replicas with cameras or that replica studies may be biased against birds by being less likely to attract predation attempts? This is an important issue and deserves a clear and logical, evidence-based argument.
L283. Your Results should provide a specific discussion of the species that interacted with replicas and those that did not to provide a basis for the discussion here (i.e., citation of Table 3 along with an overview of the findings). This is the first we have heard about ocelots being present but not interacting.
L340. As indicated above, failures of the cameras to record attacks that were documented by the replicas are important for your overall argument. I think they deserve attention earlier in the discussion rather than as a side issue in camera placement. Importantly, I did not find a clear statement in the results about these failures.
L342. I do not find the paragraph on data management relevant to your Discussion. In the Methods, you could briefly indicate how you approached this issue and why.
L351ff. Many of your contributions could also be made by still photography. I think that it would be appropriate to highlight what is gained by video in comparison to still photography, not just in comparison to the absence of any image recording.
L364-366. I agree with the reviewer that this conclusion is too far beyond the scope of your study.
Figure 1. Captions should be able to stand somewhat independently of the text. Explain briefly what kind of field studies. Provide letters for each habitat photograph and link the letters to the names of the locations.
Figure 2. This is not truly a collage any more than any other 4-panel figure. Add letters to the panels to make linking them to the species more concise. I can’t really work out the trumpeter image. Is this the head visible? I assume it is a view from behind and above. Note my suggestion above that this photo may not be needed.
Table 1. Several major and minor changes were proposed above.
Table 2. In addition to the above request for clarification, note that you may not have rigorously defined ‘trap-day’. Vertical lines should be removed. Horizontal lines are less of an issue but will probably be unnecessary. # is not a suitable abbreviation for number. The caption has too much detail. The terms should all be defined in the methods and you can simply refer to them with a statement such as ‘Numbers in parentheses indicate number of events per trap-day’.
Table 3. Follow advice above to make changes consistent with those in the other tables.
Supplementary data. Rev 1 suggested that data from one site were missing from Supp Data 1. There was no such problem in my version of the pdf, although the way that the table was structured with a separate column for study location left blank for most rows might have contributed to the impression of missing data. As to restructuring the supplementary table with your data, I agree that the present organization separate rows for the summaries of each trap are not the ideal presentation. However, I also recognize that you don’t want to provide the data that are critical to future papers, so I would accept this format as long as you provide a key to allow a reader to fully understand the table, including colors and abbreviations. If you make any changes in the terms or variables in your analysis, be sure that these are reflected in the supplementary data file.

Reviewer 1 ·

Basic reporting

I reviewed a previous version of this manuscript and was pleased to see that the authors have made improvements between iterations. I maintain that this is an excellent idea that represents a potentially useful and informative advancement on replica studies. The writing is clear and unambiguous, throughout.

However, the Introduction requires a short paragraph on the ecology of snakes and, specifically, the snake replicas used in this study, to provide much-needed background information that would inform subsequent inferences. For example, are any highly venomous? If so, this could have implications for the assessment of ‘predation’ where it is instead recontextualised as disabling a potentially dangerous snake. Further, the Discussion is poorly referenced, with few calls to existing literature.

The structure of the article is appropriate, but the Results section is, unfortunately, extremely thin.

Experimental design

The study is original and within the scope of PeerJ. The research question is acceptably defined, though there are no explicit hypotheses and the context of 'predation' is contentious (see Validity of the findings). The experimental design is appropriate, though a number of relevant details are missing from the Methodology. The novelty of the study is clear and there were no ethical issues.

Validity of the findings

No statistical analyses were carried out, despite many potential avenues of investigation. Nevertheless, the findings were broadly valid. The structure of the data provided in Supplemental Information made assessment difficult and I would recommend revision of that particular document. The Conclusions are appropriate. However, both the Discussion and Conclusions are let down by their narrow scope. There is little consideration of how this methodology could be of benefit and fill knowledge gaps beyond simple making artificial prey studies ‘better’.

I am not convinced by the author’s loose definition of ‘predators’ and ‘predation’ as it is not clear that all detected species truly constitute predators of the snakes, or replicas thereof. Further, there is no discussion of the relevance of species’ traits and ecologies and how this informs interpretations of observed behaviours. I suggest that the paper is re-cast, reconsidering the validity of statements about predators and predation.

Additional comments

Abstract

L29-30: The use of the word ‘prey’ confuses things. I suggest replacing with ‘replicas’. Further, ‘unmarked’ has no prior context in the abstract and requires explanation
.
L36: I don’t think the authors need to mention subsets in the abstract – this is a methodological note that can be brushed over, here.

L39-40: This is a broad statement and I’m not convinced that it is supported by the findings. It would be more appropriate – and accurate - to say that videography can be used to study artificial prey experiments and associated predator behaviour.

Introduction

L67: The reference can be removed from this sentence as it is invoked for each of the three noted shortcomings.

L104-108: These are methods and shouldn’t be placed here. This section also requires hypotheses, if any were generated (I’d assume that they were).

Materials and Methods

L119: Which species, specifically? Table 1 notes phenotypes but also “4 Micrurus species”. The authors should provide pictures of the real snake vs the replica in Supplemental Information to facilitate reproducibility and synthesis. What impact does using various morphs have in multi-species communities (e.g. might some be seen as a greater threat/more viable prey than others)? How many replicas were placed in each location? If this varied, why? How far away were replicas from cameras?

L132-143: Was camera placement random or according to certain features, e.g. paths? If the latter, what impact does this have on detection probability by certain species? What was the average distance (± SD) between cameras?

L144-149: Were obvious redetections also omitted? For instance, if a herd of peccary leave the area on 29 minutes and more peccaries turned up – from the same direction in which the others left and with the same number of individuals – on 31 minutes, was that counted as a new detection?

L152-153: There’s a big difference between ‘capable’ and ‘likely to’. Also, this is playing with the definition of ‘predator’. Predators prey on other species. Is a lethal interaction between a bird (where predation has not been previously observed) and a snake where the snake is not consumed or carried off truly predation? Perhaps the authors should reconsider recasting the paper as an exploration of interactions with replicas, rather than a study explicitly focused on predation.

L155: Table S1 appears to be incomplete. There is no information for Tiputini Biodiversity Station, Orellana, Ecuador.

L155-156: Why were markings left by these species omitted? The authors already use a loose definition of ‘predation’. Were any of these markings particularly damaging or potentially lethal? Was recorded behaviour indicative of precautionary or otherwise aggressive behaviour? If so, there doesn’t seem to be a good reason to leave them out of the manuscript.

Results

While the authors state that “predators were more often observed to avoid replicas…”, there has been no attempt made to quantify this statistically. There are many models that could be explored with these data, including: mammals vs birds (attacks); mammals vs birds (cameras); damaged vs undamaged; attacks vs avoidances; differences between sites; differences between replica morphs within sites; etc.

There are also more fundamental metrics and associated plots than could be described and provided. For example, what was the time-to-detection for each putative predator (and, for the Discussion, how might this be influenced by local population densities and detection probability)? How many detections of each putative predator? This is in Supplemental Information, I know, but this is relevant information that should be described here, along with bar plots (a,b,c for each site). What was the temporal pattern of interactions (perhaps a kernel density plot for each site)?

There is also an apparent discontinuity between attack counts. For example, 170 attacks were counted from clay markings compared to 6 from cameras, across 54 detections in Mexico. How were clay marks counted? Further, the authors state that gray foxes, opossums and armadillos were the species identified as interacting on camera footage while clay marks showed avian attacks. This should be acknowledged and discussed – were bird attacks simple not captured by the cameras?

L184 & 204 & 228: I don’t see the relevance of pointing out how many cameras did not detect interactions.

L195-197: Why wasn’t the area searched more thoroughly in the first instance? 4m is not a great distance.

L197-200: The field categorisation is incorrect, as-read as it seems that the replica sustained damage. There is a big difference between ‘not attacked’ and ‘we could not identify the attacker’.

L220: Redundant; delete.

Discussion

L263-270: The authors note that their findings have ‘implications’, but are there any beyond increasing statistical power? What about ecological interactions between species?

L275: Rephrase, avoiding the implicit suggestion that the bias might be trivial.

L281-282: This is unconvincing. Just because a predator appears from a different direction, that does not mean that they did not detect scent. Also, predators can follow human paths, regardless of scent.

L288: “… would [likely] increase…”

L291-293: This is based on the assumption that real, mobile animals are encountered by apparently oblivious species as frequently as immobile replicas. Further, while I agree that previously unobserved interactions between species may be important in shaping morphology, etc., it is a bit of a stretch to suggest that a species that ignores a replica is a candidate for shaping the anti-predator behavioural repertoire of the real thing.

L299-303: Combine costs and present one figure per site. Batteries and SD cards are essential, after all.

L308-312: This invites a question regarding the cost and feasibility of DNA analysis. The authors also need to provide a more thorough justification for camera traps, e.g. conservation relevance, developing new questions, etc.

L316-317: Why might this be?

L321-322: Was this done during the current study? It might be useful for the authors to provide information on the costs of their study with and without camera traps.

L324: The use of transects should be mentioned in the Methods.

L325-328: This is also detail missing from the methodology.

L330-341: This can be abbreviated by recommending field trials for appropriate camera settings.

L334: How did new captures and a full SD card cause the loss of previous data? This doesn’t make sense.

Conclusions

L364-366: I agree, but this doesn’t logically follow from the findings of this manuscript.

·

Basic reporting

No comment

Experimental design

No comment

Validity of the findings

No comment

Additional comments

Camera trap videos have been underused in predator-prey experiments and could contribute a great deal towards our understanding of how prey characteristics influence predator response. The authors use data collected as part of three experiments from Ecuador, Mexico and the USA to demonstrate the superior ability of video footage to characterize predator behavior, when compared to the traditional methodology. Overall, this manuscript is well written and topical. However, the authors have included some extraneous information, which provides context and site-specific details, but is unnecessary for delivering the overall message that the use of videography in aposematic/mimetic model experiments greatly improves the ability of investigators to study predator-prey interactions in multi-predator systems. The authors should spend some additional time editing the discussion and Table 2 with an eye toward eliminating any unnecessary details. I have included some specific comments below.

30: “unmarked prey” word choice—easily confused for mark-recapture nomenclature—perhaps something like “prey replicas that have not been damaged by predatory attacks.”
129-130: Need full names of museums—either in text or supplementary materials
163-164: Clarify
204-207: Need to some clarification of these numbers of detections/attacks reported here. There were only 54 events were predators were detected on camera, but there were 92 attacks made by mammalian predators and 78 attacks by avian predators. It is not clear how these numbers add up.
229-231: See previous comment
298-303: Discussion of the cost of cameras at scale of entire experiment is unnecessary. It would be sufficient that say that for some experimental designs the cost of cameras may be prohibitive.
317-322: Discussion of additional set-up time/transportation costs is also unnecessary.
323-341: This paragraph may also be unnecessary. While specific use-case suggestions may be helpful to some readers, these details are tangential to the main message of the manuscript.

---

## Round 0.2 · accepted · Accept

Thank you for the thorough revision of your manuscript, which I feel has greatly improved its clarity and potential usefulness to other researchers. I thoroughly enjoyed reading this version. I found only a couple of places where I could suggest minor changes in wording that would improve the manuscript. These are indicated on the attached pdf and can be corrected during the publication process.

I apologize for the delay in completing the decision process on your manuscript. I had hoped that reviewers would provide feedback on your changes. However, neither of the previous reviewers was available, so I made the decision not to send it out to new reviewers but to review the changes myself. Unfortunately, at the same time, quite a few other articles arrived that needed attention and created a backlog.

The track changes version of your manuscript was not in the normal format. It was completely highlighted in yellow rather than showing the added and deleted words that appear when using track changes in Microsoft Word. I therefore read the pdf of the new version without explicit reference to the changes. Perhaps a track changes version would not have helped much, given the extensive changes.

#